**Data Availability Statement:** All relevant data are within the paper and its Supporting Information files.

# Unique referral system contributes to long-term net clinical benefits in patients undergoing secondary prevention therapy after percutaneous coronary intervention

**Shigetaka Kageyama** [1] *, **Koichiro Murata**[1] , **Ryuzo Nawada**[1] , **Tomoya Onodera**[1] , **Yuichiro Maekawa**[2] 

1 Department of Cardiology, Shizuoka City Shizuoka Hospital, Shizuoka, Japan, 2 Internal Medicine III, Hamamatsu University School of Medicine, Shizuoka, Japan

☯ These authors contributed equally to this work.
‡ TO and YM also contributed equally to this work.
* shigenewf@yahoo.co.jp

## Abstract

Cardiovascular disease, including ischemic heart disease, is a leading cause of death worldwide. Improvement of the secondary prevention of ischemic heart disease is necessary. We established a unique referral system to connect hospitals and outpatient clinics to coordinate care between general practitioners and cardiologists. Here, we evaluated the impact and long-term benefits of our system for ischemic heart disease patients undergoing secondary prevention therapy after percutaneous coronary intervention. This single-center retrospective observational study included 3658 consecutive patients who underwent percutaneous coronary intervention at Shizuoka City Hospital between 2010 and 2019. After percutaneous coronary intervention, patients were considered conventional outpatients (conventional follow-up group) or subjected to our unique referral system (referral system group) at the attending cardiologist's discretion. To audit compliance of the treatment with the latest Japanese guidelines, we adopted a circulation-type referral system, whereby general practitioners needed to refer registered patients at least once a year, even if no cardiac events occurred. Clinical events in each patient were evaluated. Net adverse clinical events were defined as a combination of major adverse cardiac, cerebrovascular, and major bleeding events. There were 2241 and 1417 patients in the conventional follow-up and referral system groups, with mean follow-ups of 1255 and 1548 days and cumulative net adverse clinical event incidences of 27.6% and 21.5%, respectively. Kaplan–Meier analysis showed that the occurrence of net adverse clinical events was significantly lower in the referral system group than in the conventional follow-up group (log-rank: P<0.001). Univariate and multivariate analyses revealed that the unique referral system was a significant predictor of the net clinical benefits (hazard ratio: 0.56, 95% confidence interval: 0.37–0.83, P = 0.004). This result was consistent after propensity-score matching. In summary, our unique referral system contributed to long-term net clinical benefits for the secondary prevention of ischemic heart disease after percutaneous coronary intervention.

**Funding:** The author(s) received no specific funding for this work.

**Competing interests:** The authors have declared that no competing interests exist.

## Introduction

In most developed countries, cardiovascular disease, which includes ischemic heart disease (IHD), stroke, and peripheral artery disease, is and will remain the leading cause of death in men and women [1]. Further, IHD is becoming the leading cause of death worldwide. In a study of patients from 52 countries, modifiable factors accounted for >90% of the attributable risk of a first myocardial infarction (MI) [2]. Thus, secondary prevention of IHD is an important aspect of health care. There are established guidelines for patients with IHD to achieve secondary prevention [3, 4]. Follow-up visits should be to the general practitioner who may refer to a cardiologist in cases of uncertainty (class 1, level C). Nevertheless, the long-term prognosis of these patients is unknown, and these guidelines are based on trials, which have mainly been conducted in specialized centers by experienced cardiologists [5–8]. Most patients stabilized with optimal medical therapy, percutaneous coronary intervention (PCI), and coronary artery bypass grafting (CABG) are treated by general practitioners who evaluate their condition using simple examinations to prescribe medication. Because Shizuoka City has fewer cardiologists than usual (as indicated in the Japanese guidelines) [9], we established a unique referral system to connect the hospital and outpatient clinics in order to better coordinate care between general practitioners and cardiologists. When we set up the referral system, we aimed to provide optimal medical treatment for patients usually managed by general practitioners, not cardiologists, to quickly identify the signs of IHD progression. The goal was to enable diagnosis and avoid fatal cardiovascular events to achieve the secondary prevention of IHD.

Here, we aimed to evaluate the impact of our unique audit referral system and its long-term net clinical benefits to IHD patients undergoing secondary prevention therapy after PCI.

## Materials and methods

### Study design, patient recruitment, and establishment of our referral system

We included all consecutive patients who successfully underwent PCI at Shizuoka City, Shizuoka Hospital between January 2010 and October 2019. PCI was performed strictly following the latest Japanese guidelines, and physiological studies confirmed that patients did not have significant residual ischemia. Cases of PCI failure and in-hospital death at index admission were excluded from the survival analysis. In addition, events occurring before discharge were excluded. Patients who could not confirm outpatient visits after discharge whether at our hospital or with a practitioner were also excluded. The remaining patients were divided into two groups depending on whether they underwent conventional outpatient follow-up (outpatient follow-up) or were selected by their attending cardiologist to be enrolled in our unique referral system at discharge after the index event (referral system group) (details below). In this study, patients who experienced events before enrollment were classified into the conventional follow-up group.

The Shizuoka Ischemic Heart Disease registry was established in 2009 to connect cardiologists in the Shizuoka City Hospital with more than 200 general practitioners in Shizuoka City. The registry is for patients who undergo interventions or only need optimal medical therapy. This study was conducted in accordance with the Helsinki Declaration and was approved by the institutional review board (Shizuoka City Shizuoka Hospital Medical Research Ethics Review Committee 20–36). Written informed consent was obtained from the enrolled patients.

### Follow-up and data collection

To audit the treatment of general practitioners, we adopted a circulation-type cooperative form (Fig 1). General practitioners were required to refer registered patients to the department

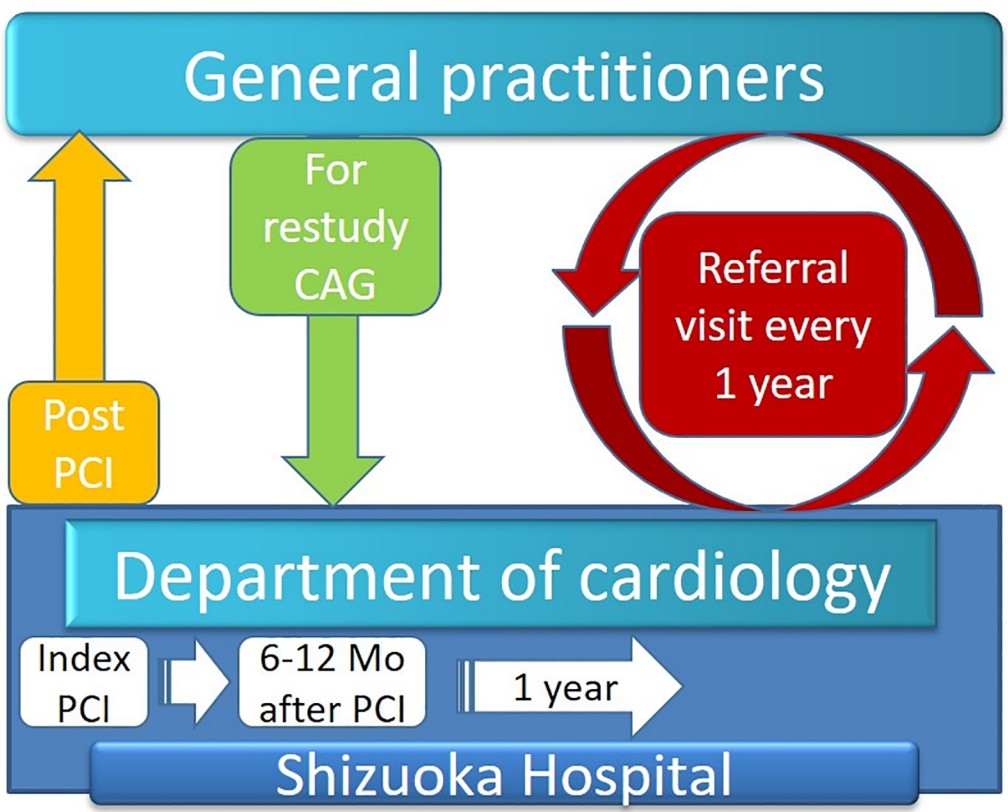

**Fig 1. Structure of our ischemic heart disease (IHD) referral system.** We established a cooperative form for circulation between general practitioners and cardiologists in our hospital. Follow-up angiography was generally performed 6–12 months after PCI. When the patients were registered in the referral system, general practitioners were required to follow up with the patients at least once a year, even in the absence of an event. IHD, ischemic heart disease.

of cardiology at least once a year, even if no cardiac events occurred. They were also requested to examine electrocardiograms every 3 months and order blood tests (including complete blood count; liver, renal function, and electrolyte tests; low-density-lipoprotein cholesterol [LDL-C]; and HbA1c) to assess risk factors every 6 months. Cardiologists audited medications and risk factors and provided advice to general practitioners via letters. The kick-off meeting was held in May 2009, and the registration process begun. Between May 2009 and April 2020, 2583 patients were enrolled in the registry. Data from yearly follow-up visits (including vital signs and findings of physical examination, electrocardiograms, chest X-rays, and laboratory tests) were collected by the investigators when patients visited the hospital. Additional follow-up information, including vital signs, mortality, additional hospitalizations, and the status of antiplatelet therapy, was collected by contacting the patients, their relatives, or referring physicians via a questionnaire or telephone calls (patients only). Data on patients who were lost to follow-up were removed from the follow-up data on the last study day.

We routinely conduct physical examinations to assess blood pressure, heart murmurs, electrocardiogram, chest X-ray, and blood tests including LDL-C and HbA1c. Medications are reviewed to determine whether they are optimal for the patient's current condition.

Conventional follow-up was performed by cardiologists in our hospital and/or by general practitioners in outpatient clinics (Fig 2). Follow-up information was obtained from the

A

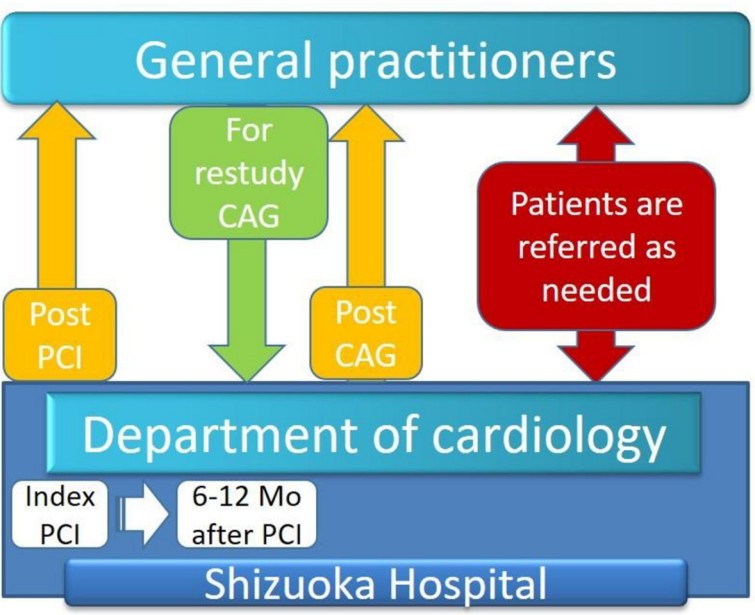

B

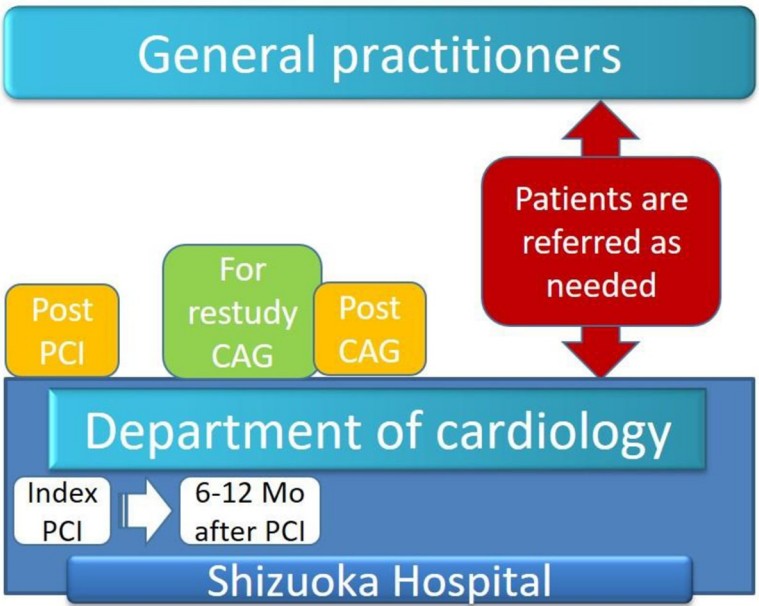

**Fig 2. Form of conventional follow-up.** In conventional follow-up, cardiologists or general practitioners continued to treat the patients with medications, and the patients were referred as needed at the discretion of the attending doctors.

medical charts, referrals from home doctors, and telephone calls to a limited number of patients. Regardless of enrollment in our referral system, patients underwent follow-up angiography in our hospital within 6 to 12 months after the index PCI unless there was a contraindication such as renal dysfunction or patient refusal.

Follow-up data were obtained when patients made regular outpatient visits to our hospital at annual referral visits following the Shizuoka IHD referral system, and at non-routine visits after referral from a general practitioner in relation to an event. In order to supplement missing death events and non-cardiac events, we also performed a prognostic study by analyzing telephone and medical records obtained between March 2019 and May 2020.

## Endpoints and definitions

Death was regarded as cardiac in origin unless obvious non-cardiac causes were identified. Any death during the index hospitalization was considered a cardiac death. Angina pectoris (AP), MI, and acute coronary syndrome (ACS) were adjudicated according to the latest guidelines from the Japanese Circulation Society, based on universal definitions [10]. Major adverse cardiac and cerebrovascular events (MACCE) were defined as a combination of cardiac death, MI, AP, ventricular arrhythmia, congestive heart failure (CHF) with or without valve disease, peripheral artery disease (PAD), and ischemic stroke. MI was classified as either type 1 or type 2 according to the Fourth Universal Definition of Myocardial Infarction [11]. AP was defined as angina necessitating hospitalization with confirmation of ischemia by angiography and/or scintigraphy. Ventricular arrhythmia was defined as ventricular tachycardia or ventricular fibrillation detected by 12-lead electrocardiography and warranting hospitalization and/or antiarrhythmic drugs. Congestive heart failure was defined as heart failure necessitating hospitalization, oxygenation, and diuretics. Peripheral artery disease was defined as arteriosclerosis obliterans necessitating intervention. Ischemic stroke was defined as stroke necessitating hospitalization and confirmed by head magnetic resonance imaging. Bleeding events were recorded as hemorrhagic disorders when any antiplatelet drug was discontinued or patients required hospitalization or blood transfusions. Major or minor bleeding was defined according to the Thrombolysis in Myocardial Infarction Trial bleeding criteria [12]. Net adverse clinical events (NACE) were defined as a combination of MACCE and major bleeding events. The primary endpoint of this study was NACE occurrence.

## Management of risk factors

The recommended antiplatelet regimen after PCI included aspirin (100 mg daily) indefinitely and thienopyridine (200 mg ticlopidine daily, 75 mg clopidogrel daily, or 3.75 mg prasugrel daily) for at least 6 months [9]. The dual antiplatelet therapy (DAPT) duration was decided at the discretion of each cardiologist. We used the latest Japanese guidelines for the secondary prevention of IHD to set a target value as a threshold for starting medications to control risk factors [13]. Statins were recommended if the low-density-lipoprotein cholesterol level was >100 mg/dl. Improvement in diet or the addition of oral hypoglycemic drugs was recommended if the glycated hemoglobin level was >7%. Antihypertensive drugs were recommended if the systolic blood pressure was >130 mmHg and/or the diastolic blood pressure was >80 mmHg when measured at home.

## Statistical analyses

Categorical variables are expressed as numbers and percentages, and continuous variables as mean ± standard deviation, unless otherwise indicated. Patient characteristics were compared between the two groups using unpaired t-tests for continuous variables and Fisher's exact test for categorical variables. Multivariate analysis was performed using the Cox regression analysis for variables with P<0.10 in the univariate analysis. Cumulative incidences were calculated using the Kaplan–Meier method and log-rank analysis. Propensity-score matching was performed after raw data analysis to correct significant background factors contributing to group

differences. The follow-up term was estimated to be significantly longer in the conventional follow-up group than in the unique referral system group. The start of conventional follow-up was within 90 days after the index PCI. Therefore, for propensity score matching, we only included patients from the unique referral group who had less than 100 days between the index PCI and registration. We matched the factors with P<0.1 in the baseline comparison between the two groups. All statistical analyses were performed using R (The R Foundation for Statistical Computing, Vienna, Austria). All reported P-values were two-sided, and P<0.05 was considered statistically significant. Significant predictors of clinical events were presented with odds ratios (OR) and 95% confidence intervals (CI).

## Results

### Baseline characteristics

A total of 3739 patients who underwent PCI between January 2010 and October 2019 were enrolled. In-hospital death at index admission occurred in 81 cases. The remaining 3658 patients included 2241 patients in the conventional follow-up group and 1417 who were enrolled in the Shizuoka IHD referral system; the mean ages were 69.9 and 68.6 years (P<0.001) while the number of men was 1679 (74.9%) and 1097 (77.4%), respectively. In the conventional follow-up and referral system groups, 33.6% and 31.4% patients, respectively, underwent emergency PCI for ACS as an index event; furthermore, 33.5% and 31.4% patients, respectively, were diagnosed with diabetes at the time of registration. Complete baseline patient characteristics are presented in Table 1.

Medication use after index PCI was as follows: antiplatelet therapy: 98%, anticoagulation therapy: 8.9%, calcium channel blockers: 55%, angiotensin-converting enzyme inhibitors and/ or angiotensin II receptor blockers: 60.9%, beta-blockers: 24.1%, and statins: 75.7%. There were no significant differences between the groups regarding medication use. Data regarding current medications were obtained only from the Shizuoka IHD referral system group, and the rates of statin prescription and anticoagulation therapy became significantly higher than the baseline rates (90% [P<0.001] and 11.7% [P = 0.024], respectively).

Details regarding risk factor control were available only for the Shizuoka IHD referral system group. Risk factor control at registration showed the following results: systolic blood pressure, 126.6 ± 17.5 mmHg; diastolic blood pressure, 69.5 ± 12.6 mmHg; LDL-C, 93 ± 28.5 mg/ dL; and HbA1c (in patients with diabetes), 6.7% ± 1.0%. The average LDL-C level at the latest follow-up was significantly decreased relative to the baseline value (93 ± 28.5 to 88.0 ± 21.3 mg/dL; P < 0.001). HbA1c control in patients with diabetes significantly worsened (6.7% ± 1.0% to 7.0% ± 1.1%; P < 0.001), while there was a significant increase in both the systolic (126.6 ± 17.5 mmHg to 133.7 ± 16.5 mmHg; P < 0.001) and diastolic (69.5 ± 12.6 mmHg to 75.0 ± 11.8 mmHg; P < 0.001) blood pressure values from baseline to the latest follow-up.

### Incidences of clinical events

The mean follow-up term differed significantly between the conventional follow-up and referral system groups (1255 ± 1089 versus 1548 ± 1067 days, P<0.001). Cumulative incidences of all-cause death were 11.3% and 6.6% in the conventional follow-up and referral system groups, respectively (P<0.001). Cardiovascular death occurred in 3.7% and 1.3% patients (P<0.001), while MACCE occurred in 24.6% and 19.2%, respectively (P<0.001). Revascularization accounted for 65.4% of all MACCE (n = 538). CHF accounted for 25.9% (n = 213). MI accounted for 6.5% (n = 54). Ventricular arrhythmia accounted for 2.9% (n = 24). Each factor did not have any statistically significant differences between the conventional follow-up group and the referral system group. Major bleeding events occurred in 4.6% and 3.0% patients in

**Table 1. Baseline patient characteristics.**

| Factor | | Referral system | | P value |
|---|---|---|---|---|
| | | Conventional | IHD registry | |
| Patients number | | 2241 | 1417 | |
| Age, years | | 69.89 ± 11.52 | 68.61 ± 11.18 | 0.001** |
| Male sex (%) | | 1679 (74.9) | 1097 (77.4) | 0.088 |
| BMI, kg/m$^2$ | | 23.90 ± 5.48 | 24.72 ± 20.33 | 0.07 |
| Comorbidities | | | | |
| Smoking status (%) | Never | 803 (35.8) | 521 (36.8) | 0.791 |
| | Prior | 1057 (47.2) | 665 (46.9) | |
| | Current | 381 (17.0) | 231 (16.3) | |
| Hypertension (%) | | 1519 (67.8) | 940 (66.3) | 0.366 |
| Systolic BP, mmHg | | 127.67 ± 22.05 | 130.79 ± 22.84 | <0.001*** |
| Diastolic BP, mmHg | | 74.98 ± 77.88 | 74.36 ± 15.83 | 0.773 |
| Heart rate, bpm | | 74.63 ± 64.86 | 70.81 ± 23.03 | 0.394 |
| CKD (%) | | 542 (24.2) | 242 (17.1) | <0.001*** |
| Hemodialysis (%) | | 25 (1.8) | 17 (2.4) | 0.327 |
| Ccr, mL/min | | 66.66 ± 33.82 | 71.14 ± 30.63 | <0.001*** |
| eGFR, mL/min/1.73 m$^2$ | | 60.29 ± 22.45 | 63.77 ± 20.11 | <0.001*** |
| Dyslipidemia (%) | | 1069 (47.7) | 682 (48.1) | 0.812 |
| Diabetes (%) | | 751 (33.5) | 445 (31.4) | 0.193 |
| Atrial fibrillation (%) | | 117 (8.3) | 46 (6.5) | 0.143 |
| Prior MI (%) | | 403 (18.0) | 231 (16.3) | 0.194 |
| Prior CHF (%) | | 274 (12.2) | 93 (6.6) | <0.001*** |
| Prior stroke (%) | | 186 (8.3) | 65 (4.6) | <0.001*** |
| Prior PAD (%) | | 136 (6.1) | 54 (3.8) | 0.003** |
| Prior PCI (%) | | 452 (20.2) | 327 (23.1) | 0.038* |
| Prior CABG (%) | | 153 (6.8) | 66 (4.7) | 0.008** |
| Index PCI characteristics | | | | |
| ACS (%) | | 754 (33.6) | 483 (34.1) | 0.774 |
| LM, LAD lesion (%) | | 1463 (65.3) | 898 (63.4) | 0.242 |
| Type B2/C lesion (%) | | 907 (52.8) | 600 (55.1) | 0.244 |
| Multiple target vessels | | 290 (17.9) | 210 (19.4) | 0.538 |
| LVEF (%) | | 49.51 ± 10.91 | 50.71 ± 10.17 | 0.043* |
| Lesion length, mm | | 11.40 ± 8.00 | 11.42 ± 7.09 | 0.926 |
| Reference diameter, mm | | 2.53 ± 0.68 | 2.59 ± 0.66 | 0.011* |
| Post DS (%) | | 8.70 ± 6.74 | 8.81 ± 6.80 | 0.648 |
| Post MLD, mm | | 2.81 ± 0.52 | 2.83 ± 0.52 | 0.295 |

Values are mean ± standard deviation.

*P<0.05

**P<0.01, and

***P<0.001.

Abbreviations: IHD, ischemic heart disease; BMI, body mass index; BP, blood pressure; CKD, chronic kidney disease; Ccr, creatinine clearance; eGFR, estimated glomerular filtration rate; MI, myocardial infarction; CHF, congestive heart failure; PAD, peripheral artery disease, CABG, coronary artery bypass grafting; PCI, percutaneous coronary intervention; ACS, acute coronary syndrome; LM/LAD, left main and/or left anterior descending artery; LVEF, left ventricular ejection fraction; DS, diameter stenosis; MLD, minimum lumen diameter.

**Table 2. Clinical events during follow-up.**

| Factor | Referral system | | P value |
|---|---|---|---|
| | Conventional | IHD registry | |
| Number of patients | 2241 | 1417 | |
| All-cause death (%) | 254 (11.3) | 93 (6.6) | <0.001*** |
| CV death (%) | 82 (3.7) | 19 (1.3) | <0.001*** |
| MACCE (%) | 551 (24.6) | 272 (19.2) | <0.001*** |
| Major bleeding (%) | 104 (4.6) | 43 (3.0) | 0.016* |
| Net clinical event (%) | 618 (27.6) | 304 (21.5) | <0.001*** |
| Follow-up term | 1254.64 ± 1089.29 | 1548.12 ± 1066.60 | <0.001*** |

Values are mean ± standard deviation.

*P<0.05

**P<0.01, and

***P<0.001.

Abbreviations: IHD, ischemic heart disease; CV, cardiovascular; MACCE, major adverse cardiac and cerebrovascular event.

the conventional follow-up and referral system groups, respectively (P = 0.016). Net clinical events were observed in 27.6% and 21.5% patients in the conventional follow-up and referral system groups, respectively (P<0.001, Table 2).

## Predictors of the primary endpoint

Predictors of the primary endpoint were verified among 3739 patients, and 994 primary endpoints were detected. The age at baseline was significantly higher in the NACE group than in the event-free group (71.2 ± 10.8 vs. 68.9 ± 11.6, P<0.001), while the left ventricular ejection fraction (LVEF) was significantly lower (47.7 ± 11.7 vs. 50.5 ± 10.4%, P<0.001). The incidences of hypertension (HT), diabetes, chronic kidney disease (CKD), and PAD and prior histories of MI, CHF, stroke, PCI, and CABG were significantly higher in the NACE group than in the event-free group. The rate of inclusion in the unique referral system was significantly lower in the NACE group than in the event-free group (33.0% vs. 40.7%, respectively; P<0.001). Smoking habits significantly differed between the groups (P = 0.045).

According to the multivariate analysis, a higher LVEF and inclusion in the unique referral system were significant predictors of net clinical benefits (OR: 0.97, 95% CI: 0.96–0.997 and OR: 0.56, 95% CI: 0.37–0.83, respectively; Table 3).

Kaplan–Meier analysis was performed for comparing the primary endpoints between the groups. The log-rank test revealed a significant difference between the groups (P<0.001, Fig 3).

## Propensity-score matching analysis for the primary endpoint

The attending physician decided whether patients should be enrolled in the referral system; we found some significant differences between the referral system and conventional follow-up groups in terms of age, systolic blood pressure, LVEF, the rate of CKD, and prior history of CHF, stroke, PAD, PCI, and CABG (Table 1). We performed propensity-score matching and extracted 436 matched pairs. Baseline characteristics are shown in Table 4. Univariate and multivariate analyses revealed that lower systolic blood pressure and enrollment in the unique referral system were significant predictors of net clinical benefits (OR: 1.01, 95% CI: 1.00–1.02 and OR: 0.48, 95% CI: 0.28–0.81, respectively). Other factors that also predicted net clinical

**Table 3. Predictors of net clinical benefits.**

| Factor | Net adverse clinical events | | P value | Multivariate analysis | | |
|---|---|---|---|---|---|---|
| | Yes (n = 994) | No (n = 2745) | | HR | 95% CI | P value |
| Age, year | 71.21 ± 10.77 | 68.92 ± 11.62 | <0.001 | 1.007 | 0.99–1.03 | 0.413 |
| Smoking (%) | 619 (62.3) | 1747 (63.7) | 0.045 | 1.13 | 0.86–1.48 | 0.375 |
| Hypertension (%) | 710 (71.4) | 1799 (65.5) | 0.001 | 1.093 | 0.75–1.59 | 0.638 |
| CKD (%) | 295 (29.7) | 533 (19.4) | <0.001 | 1.486 | 0.90–2.46 | 0.123 |
| Diabetes (%) | 362 (36.4) | 864 (31.5) | 0.005 | 0.917 | 0.61–1.37 | 0.675 |
| Atrial fibrillation (%) | 46 (9.6) | 120 (7.1) | 0.08 | 1.456 | 0.63–3.38 | 0.383 |
| Prior MI (%) | 207 (20.8) | 444 (16.2) | 0.001 | 0.921 | 0.51–1.65 | 0.782 |
| Prior CHF (%) | 138 (13.9) | 245 (8.9) | <0.001 | 1.609 | 0.77–3.35 | 0.204 |
| Prior stroke (%) | 102 (10.3) | 159 (5.8) | <0.001 | 0.869 | 0.31–2.41 | 0.788 |
| Prior PAD (%) | 81 (8.1) | 121 (4.4) | <0.001 | 1.484 | 0.59–3.73 | 0.402 |
| LVEF, % | 47.69 ± 11.68 | 50.48 ± 10.37 | <0.001 | 0.98 | 0.96–0.997 | 0.019* |
| Enter Shizuoka IHD referral system (%) | 304 (33) | 1113 (40.7) | <0.001 | 0.556 | 0.37–0.83 | 0.004** |

Values are mean ± standard deviation.

*P<0.05

**P<0.01, and

***P<0.001.

Abbreviations: HR, hazard ratio; CI, confidence interval; CKD, chronic kidney disease; Ccr, creatinine clearance; eGFR, estimated glomerular filtration rate; MI, myocardial infarction; CHF, congestive heart failure; PAD, peripheral artery disease, LVEF, left ventricular ejection fraction; IHD, ischemic heart disease.

benefits included an absence of chronic obstructive pulmonary disease and anticoagulation therapy and a higher estimated glomerular filtration rate (Table 5).

Kaplan–Meier analysis was performed to compare the incidence of NACE between the referral system and conventional follow-up groups. The log-rank test revealed a significant difference between the groups (P = 0.007, Fig 4).

## Discussion

The main finding in this study was that enrollment in the unique referral system after PCI was the strongest predictor for net clinical benefits, even after including factors such as age, past medical histories, coexisting diseases, and lesion characteristics. This finding was consistent after a matched pair analysis. The latest guidelines for chronic coronary syndrome recommend a periodic visit to cardiovascular healthcare professionals to reassess any potential changes in the risk status of patients, clinical evaluations with lifestyle-modification measures, adherence to targets of cardiovascular risk factors, and development of comorbidities that may affect treatments and outcomes (Class 1, Level C) [4]. Several attempts have been made to improve patient prognosis through the introduction of standardized secondary prevention methods. A systematic review revealed that disease management programs improve the quality of care, reduce admissions to hospital, and enhance the quality of life or functional status [14]. Furthermore, Gong et al. reported that among cardiologists in the university hospital, standardized follow-up helped improve secondary prevention of coronary heart disease [15].

Recent outcomes of secondary prevention after an IHD event with conventional optimal medical therapy were reported in several international multicenter prospective randomized studies. When administering evolocumab to patients with stable cardiovascular disease, the cumulative incidence of clinical outcomes was 9.8%, while that of ischemic stroke was 1.2% and that of hemorrhagic stroke was 0.21%, during a 2.2-year follow-up [16]. After a 39-month

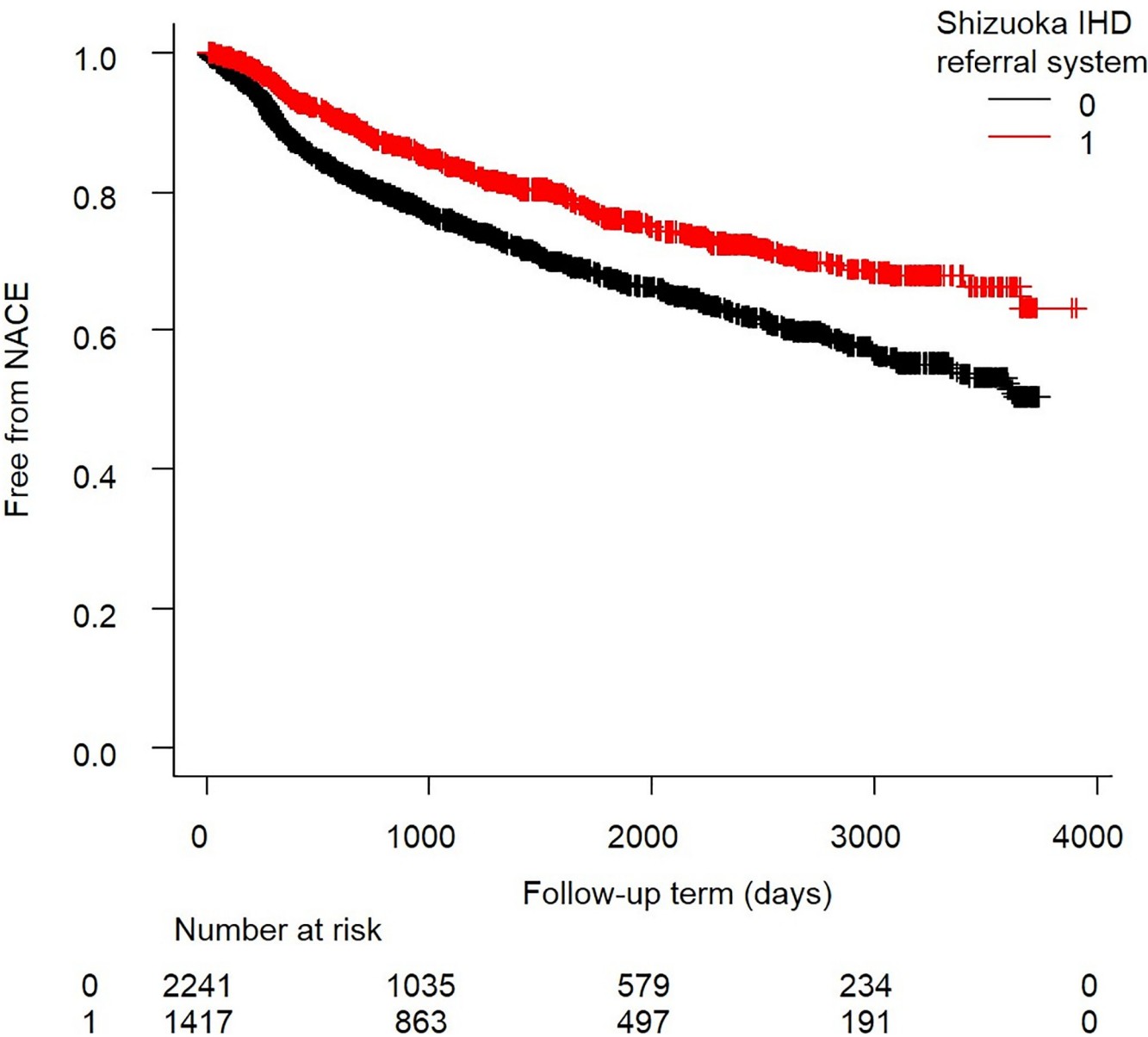

**Fig 3. Kaplan–Meier analysis of net adverse clinical events (NACE) with or without use of our unique referral system.** The red line represents the referral system group and the black line represents the conventional follow-up group. There is a significant net clinical benefit for the unique referral system group (p<0.001; log-rank test).

follow-up of patients with or without complete revascularization for acute MI, cardiovascular death, MI, ischemia-driven revascularization, unstable angina, or CHF occurred in 17.3% patients, while major bleeding events occurred in 2.5% [17]. An observational study based on data from 2199 patients with stable coronary heart disease in Copenhagen revealed a 6-year mortality rate of 20.1% [18]. Compared to studies in which heart diseases were treated by cardiologists, the real-world IHD secondary prevention after PCI in Shizuoka City was associated with more clinical events in the present study. This was due to routine restudy coronary angiography conducted 6 months to 1 year after PCI, which revealed in-stent restenosis and new lesion stenosis without symptoms. However, when compared with the conventional optimal

**Table 4. Baseline characteristics (matched pair analysis).**

| Factor | | | Referral system | | P value |
|---|---|---|---|---|---|
| | | | Conventional | IHD registry | |
| Number of patients | | | 436 | 436 | |
| Age, years | | | 66.20 ± 10.85 | 66.26 ± 10.85 | 0.935 |
| Male sex (%) | | | 341 (78.2) | 340 (78.0) | 1 |
| BMI, kg/m$^2$ | | | 24.16 ± 3.4 | 24.62 ± 9.6 | 0.354 |
| **Comorbidities** | | | | | |
| Smoking Status (%) | | Never | 173 (39.7) | 145 (33.3) | 0.022* |
| | | Prior | 167 (38.3) | 161 (36.9) | |
| | | Current | 96 (22) | 130 (29.8) | |
| Hypertension (%) | | | 254 (58.3) | 255 (58.5) | 1 |
| Systolic BP, mmHg | | | 135.46 ± 22.81 | 130.98 ± 23.97 | 0.006** |
| Diastolic BP, mmHg | | | 80.38 ± 16.76 | 78.64 ± 18.09 | 0.156 |
| Heart rate, bpm | | | 76.36 ± 36.46 | 73.91 ± 16.13 | 0.216 |
| CKD (%) | | | 20 (4.6) | 20 (4.6) | 1 |
| Hemodialysis (%) | | | 4 (2.1) | 3 (1.2) | 0.704 |
| Ccr, mL/min | | | 82.34 ± 29.19 | 81.63 ± 32.91 | 0.75 |
| eGFR, mL/min/1.73m$^2$ | | | 71.89 ± 18.29 | 70.05 ± 18.42 | 0.16 |
| Dyslipidemia (%) | | | 121 (27.8) | 146 (33.5) | 0.078 |
| Diabetes (%) | | | 97 (22.2) | 111 (25.5) | 0.302 |
| Atrial fibrillation (%) | | | 3 (1.6) | 8 (3.3) | 0.361 |
| Prior MI (%) | | | 55 (18.0) | 60 (13.8) | 0.689 |
| Prior CHF (%) | | | 3 (0.7) | 3 (0.7) | 1 |
| Prior stroke (%) | | | 5 (1.1) | 5 (1.1) | 1 |
| Prior PAD (%) | | | 1 (0.2) | 1 (0.2) | 1 |
| Prior PCI (%) | | | 55 (12.6) | 61 (14.0) | 0.618 |
| Prior CABG (%) | | | 4 (0.9) | 7 (1.6) | 0.546 |
| **Index PCI characteristics** | | | | | |
| ACS (%) | | | 335 (77) | 319 (73.2) | 0.21 |
| LM, LAD lesion (%) | | | 308 (70.6) | 317 (72.7) | 0.548 |
| LVEF, % | | | 50.80 ± 9.64 | 50.79 ± 9.62 | 0.986 |

Plus-minus values are means ± standard deviation.

*P<0.05

**P<0.01, and

***P<0.001.

Abbreviations: IHD, ischemic heart disease; BMI, body mass index; BP, blood pressure; CKD, chronic kidney disease; Ccr, creatinine clearance; eGFR, estimated glomerular filtration rate; MI, myocardial infarction; CHF, congestive heart failure; PAD, peripheral artery disease, CABG, coronary artery bypass grafting; PCI, percutaneous coronary intervention; ACS, acute coronary syndrome; LM/LAD, left main and/or left anterior descending artery; LVEF, left ventricular ejection fraction; DS, diameter stenosis.

medication treatment groups studied in recent large-scale clinical trials, our referral system group showed the same extent of improvement in net clinical events [13–18].

In this study, risk factor control in the referral group only showed an improvement in LDL-C levels and an exacerbation in HbA1c and blood pressure control compared to baseline. Despite the worsening of the latter two factors, the incidence of clinical events was still lower in the referral group than in the conventional follow-up group. This discrepancy can be explained by the fact that our unique referral system lowered the threshold for general

**Table 5. Predictors of net clinical benefits after matched pair analysis.**

| Factor | Net adverse clinical event | | P value | Multivariate analysis | | |
|---|---|---|---|---|---|---|
| | Yes (n = 168) | No (n = 702) | | HR | 95% CI | P value |
| Male sex (%) | 140 (83.3) | 541 (77.1) | 0.078 | 1.308 | 0.69–2.47 | 0.407 |
| Systolic BP, mmHg | 136.2 ± 25.6 | 132.5 ± 23 | 0.076 | 1.011 | 1.00–1.02 | 0.037* |
| eGFR, mL/min/1.73 m$^2$ | 67.69 ± 15.92 | 71.77 ± 18.83 | 0.014 | 0.987 | 0.97–1.00 | 0.076 |
| Atrial fibrillation (%) | 4 (5.7) | 6 (1.6) | 0.059 | 2.784 | 0.99–7.85 | 0.053 |
| Enter Shizuoka IHD referral system (%) | 74 (44) | 361 (51.4) | 0.053 | 0.478 | 0.28–0.81 | 0.006** |

Values are means ± standard deviation.

*P<0.05

**P<0.01, and

***P<0.001.

Abbreviations: HR, hazard ratio; CI, confidence interval; BP, blood pressure; eGFR, estimated glomerular filtration rate; IHD, ischemic heart disease.

practitioners in recommending their patients to consult with a cardiologist. In turn, this could have potentially led to a net clinical benefit for the patient since it was easier to consult with other specialists in the general hospital, such as cerebrovascular disease and bleeding event specialists, using the referral to the cardiologist as a starting point. Another potential reason is the fact that these factors perhaps do not pose the same level of risk of developing a clinical event. Many large clinical trials have reported that cardiovascular events are directly correlated with LDL-C levels in the secondary prevention of ischemic heart disease [4, 16]; however, it is possible that HbA1c levels or blood pressure are inferior risk indicators for cardiovascular events compared to LDL-C levels [19]. In a large population-based prospective study in Japan, the risk of coronary artery disease was significantly higher in the hypertensive group than in the normotensive group [20]. However, the risk probably could not have counteracted the impact of the improved prognostic effect of lower LDL-C in this study population.

Balancing ischemic and bleeding risks and minimizing complex events are the primary goals of secondary prevention. In the propensity-matched cohorts from multicenter trials conducted after PCI, patients with a subsequent ACS admission had an increased risk of mortality (hazard ratio: 4.73; P = 0.015), whereas those with unplanned revascularization did not have a significantly higher risk [21]. Conversely, discontinuation of antiplatelet therapy possibly caused new cardiovascular events, including stent thrombosis. These secondary events worsened the survival rate of the major bleeding group [22]. We previously revealed that MI, ventricular arrhythmia, and intracranial hemorrhage after registration to the referral system were significant predictors of all-cause death, while major bleeding events were a stronger predictor than coronary events [23]. The DAPT score is a novel decision tool that was recently developed to identify those more likely to experience benefits, rather than harm, from long-term therapy among patients eligible for long-term dual antiplatelet therapy [24]. In connection with the recent global definition of high bleeding risk, the latest guidelines suggest the option of short DAPT and oral anticoagulants alone after PCI [25–27]. By introducing a unique referral system, we believe that the patient prognosis was strongly improved through a holistic approach optimized for each patient, which included the auditing of medications and risk factors, preventing the occurrence of MI events by specialized tests, introducing the device, providing lifestyle guidance and family awareness, discovery, bridging the treatments of non-cardiac diseases, and utilizing the characteristics of general hospitals.

There are some limitations in this study. First, a major limitation was patient selection. Cardiologists in the hospitals selected and registered patients who complied with the referral

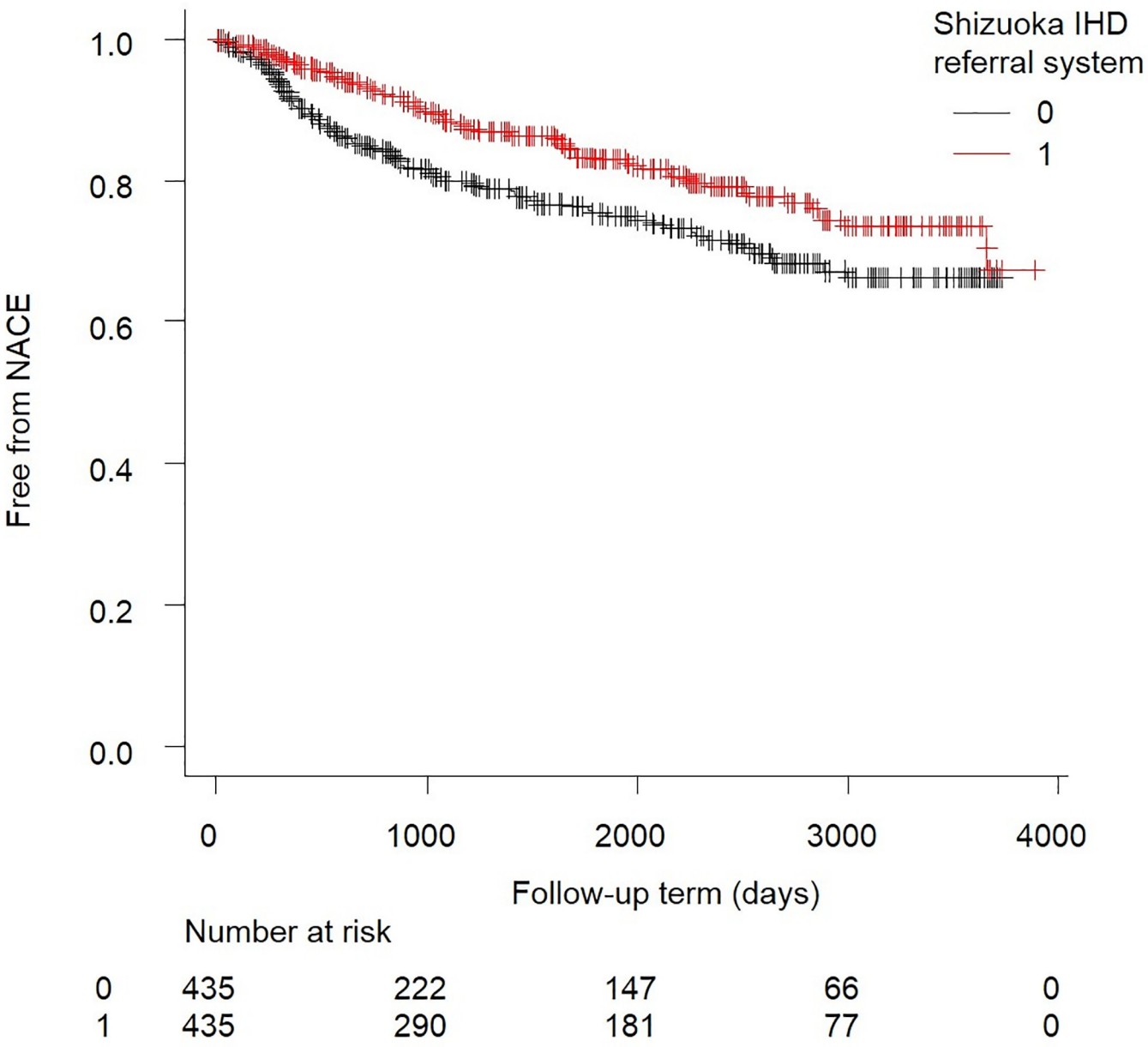

**Fig 4. Kaplan–Meier analysis of net adverse clinical events (NACE) with or without use of our unique referral system (matched pair analysis).** The red line represents the referral system group and the black line represents the conventional follow-up group. There is a significant net clinical benefit for the unique referral system group (p = 0.007; log-rank test).

system. The cardiologists tended to follow-up patients who were at "high risk" with a low LVEF and/or recurrent cardiovascular disease. Conversely, attending cardiologists tended to refer to "low risk" patients to general practitioners without using the unique referral system. For these reasons, patient characteristics of this study became heterogeneous. However, we performed matched-pair analysis to overcome the bias; this analysis generated consistent results. Second, among patients who were followed up without a referral, data on current medications were not available. Because the conventional follow-up group had a shorter follow-up period, we could not evaluate the endpoints occurring after follow-up. However, the main results will not be overturned as more events are already occurring in a short period in the

conventional follow-up group. Even after propensity score matching, some risk factors were statistically significant. A future prospective, multicenter study including more participants and different lesions and countries would be needed to confirm the utility of this unique referral system.

## Conclusions

Enrollment in our unique referral system in Shizuoka City was the strongest predictor of net clinical benefits for secondary prevention after PCI. The design of this unique referral system is useful to reduce NACE and can be standardized for clinical practice.

## Supporting information

**S1 File.**
(XLSX)

**S2 File.**
(XLSX)

## Acknowledgments

We thank all general practitioners in Shizuoka City who treated the participating patients in the current study. We also thank the cardiologists and outpatient staff who contributed to this study by assessing treatment strategies and collecting data.

## Author Contributions

**Conceptualization:** Shigetaka Kageyama, Koichiro Murata, Ryuzo Nawada, Tomoya Onodera.

**Data curation:** Shigetaka Kageyama, Koichiro Murata, Ryuzo Nawada, Tomoya Onodera.

**Formal analysis:** Shigetaka Kageyama.

**Investigation:** Shigetaka Kageyama.

**Methodology:** Shigetaka Kageyama, Koichiro Murata, Ryuzo Nawada, Yuichiro Maekawa.

**Project administration:** Shigetaka Kageyama.

**Supervision:** Tomoya Onodera, Yuichiro Maekawa.

**Validation:** Shigetaka Kageyama.

**Visualization:** Shigetaka Kageyama.

**Writing – original draft:** Shigetaka Kageyama.

**Writing – review & editing:** Shigetaka Kageyama, Tomoya Onodera, Yuichiro Maekawa.

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
