## [Decision Letter · Decision Letter 0]

7 Sep 2020

PONE-D-20-25225

Unique referral system contributes to long-term net clinical benefits in patients undergoing secondary prevention therapy after percutaneous coronary intervention

PLOS ONE

Dear Dr. KAGEYAMA,

Thank you for submitting your manuscript to PLOS ONE. After careful consideration, we feel that it has merit but does not fully meet PLOS ONE’s publication criteria as it currently stands. Therefore, we invite you to submit a revised version of the manuscript that addresses the points raised during the review process.

We look forward to receiving your revised manuscript.

Kind regards,

Yoshihiro Fukumoto

Academic Editor

PLOS ONE

Journal Requirements:

Reviewers' comments:

Reviewer's Responses to Questions

**Comments to the Author**

1. Is the manuscript technically sound, and do the data support the conclusions?

Reviewer #1: Yes

Reviewer #2: Yes

2. Has the statistical analysis been performed appropriately and rigorously? 

Reviewer #1: Yes

Reviewer #2: I Don't Know

3. Have the authors made all data underlying the findings in their manuscript fully available?

Reviewer #1: Yes

Reviewer #2: No

4. Is the manuscript presented in an intelligible fashion and written in standard English?

Reviewer #1: Yes

Reviewer #2: Yes

5. Review Comments to the Author

Reviewer #1: This single-center, retrospective observational study evaluated the impact and long-term benefits of unique referral system to 3658 consecutive patients with ischemic heart disease undergoing secondary prevention therapy after PCI. The authors showed that the occurrence of net adverse clinical events was significantly lower in the referral system group than in the conventional follow-up group even after propensity-score matching. This study is interesting and provides useful information although this is a single-center, retrospective observational study. However, this study needs to be improved in some points.

1. As the authors stated in the limitation section of the discussion, baseline patient characteristics are too heterogeneous. Even after propensity-score matching, some risk factors had statistically significant. The authors should discuss the significance of the multicenter, prospective study in this topic.

2. Regarding the Figures 3 and 4, when was the start of follow up (ie, 0 day). The authors should set the start date of the follow up to the registration date not to PCI date. The Kaplan-Meier curve demonstrated that the event rate between 2 groups seems to have the difference within 1 years. It might be a vias if the event rate has the significant difference before the registration. The authors should clearly state this point in the Method and Result sections.

Reviewer #2: This manuscript investigated clinical benefits of the audit referral system which has been built for coordinate care between general practitioners and cardiologists.

The authors found that their referral system contributed to better long-term clinical benefits for the secondary prevention of ischemic heart disease after percutaneous coronary intervention than conventional follow-up.

This reviewer has the following comments:

- Please provide the detail of MACCE.

If the majority of the overall events are revascularization, more detailed information on lesion background (e.g. complexity) and procedural information at the index PCI should be provided.

- Definition of each item in MACCE should be explained, such as MI, AP, ventricular arrhythmia, congestive heart failure, and peripheral artery disease.

- To confirm the advantages of the referral system over conventional follow-up, the data regarding risk factor control during the follow-up period would be helpful.

- The study design is unclear. Particularly, methodology for data collection should be explained clearly. Data were collected every year or at the specific time point?

6. PLOS authors have the option to publish the peer review history of their article (what does this mean?). If published, this will include your full peer review and any attached files.

Reviewer #1: No

Reviewer #2: No

---

## [Author Response · Author response to Decision Letter 0]

4 Oct 2020

5. Review Comments to the Author

Reviewer #1: This single-center, retrospective observational study evaluated the impact and long-term benefits of unique referral system to 3658 consecutive patients with ischemic heart disease undergoing secondary prevention therapy after PCI. The authors showed that the occurrence of net adverse clinical events was significantly lower in the referral system group than in the conventional follow-up group even after propensity-score matching. This study is interesting and provides useful information although this is a single-center, retrospective observational study. However, this study needs to be improved in some points.

1. As the authors stated in the limitation section of the discussion, baseline patient characteristics are too heterogeneous. Even after propensity-score matching, some risk factors had statistically significant. The authors should discuss the significance of the multicenter, prospective study in this topic.

Thank you for this important suggestion. I have added the importance of a future multicenter prospective study to the limitations paragraph of the Discussion section (lines 371-372).

2. Regarding the Figures 3 and 4, when was the start of follow up (ie, 0 day). The authors should set the start date of the follow up to the registration date not to PCI date. The Kaplan-Meier curve demonstrated that the event rate between 2 groups seems to have the difference within 1 years. It might be a vias if the event rate has the significant difference before the registration. The authors should clearly state this point in the Method and Result sections.

Thank you for your valuable input. The time between the latest PCI and enrollment in the unique referral system varied among cases, and in actual practice, some patients can be enrolled in the referral system after the event. In this study, however, patients who experienced events before enrollment were classified into the conventional follow-up group. Therefore, the event rate before registration (and after the index PCI) was not known for either group. However, the follow-up term was significantly longer in the conventional follow-up group than in the unique referral system group. The median duration from index PCI to registration in the unique referral system group was 60 days (IQR 2-299). The start of conventional follow-up was within 90 days after the index PCI. Therefore, for propensity score matching, we only included patients from the unique referral group who had less than 100 days between the index PCI and the registration. We also excluded events occurring before discharge. These aspects have been clarified in the Materials and Methods section (lines 85, 90-91, 181-185).

Reviewer #2: This manuscript investigated clinical benefits of the audit referral system which has been built for coordinate care between general practitioners and cardiologists.

The authors found that their referral system contributed to better long-term clinical benefits for the secondary prevention of ischemic heart disease after percutaneous coronary intervention than conventional follow-up.

This reviewer has the following comments:

- Please provide the detail of MACCE.

If the majority of the overall events are revascularization, more detailed information on lesion background (e.g. complexity) and procedural information at the index PCI should be provided.

Thank you for your suggestion. Revascularization accounted for 65.4% of MACCE (n = 538), and CHF accounted for 25.9% (n = 213). MI accounted for 6.5% (n = 54). Ventricular arrhythmia accounted for 2.9% (n = 24). Each factor did not have any statistically significant differences between the conventional follow-up group and the referral system group. Regarding the lesion complexity at the index PCI, the incidence of type B2/C lesions was 53.3%. The average number of target lesions was 1.22. We have added all relevant details to Table 1 and the Results section (lines 234–237).

- Definition of each item in MACCE should be explained, such as MI, AP, ventricular arrhythmia, congestive heart failure, and peripheral artery disease.

Thank you for your suggestion. I have added the relevant definitions in the Materials and Methods section (lines 149-157). MI was classified as type 1 and type 2 MI according to the Fourth Universal Definition of Myocardial Infarction (Thygesen K, Alpert JS, Jaffe AS, Chaitman BR, Bax JJ, Morrow DA, White HD; Executive Group on behalf of the Joint European Society of Cardiology (ESC)/American College of Cardiology (ACC)/American Heart Association (AHA)/World Heart Federation (WHF) Task Force for the Universal Definition of Myocardial Infarction. Fourth Universal Definition of Myocardial Infarction (2018). J Am Coll Cardiol. 2018 Oct 30;72(18):2231-2264). AP was defined as angina necessitating hospitalization, with confirmation of ischemia by angiography and/or scintigraphy. Ventricular arrhythmia was defined as ventricular tachycardia or ventricular fibrillation detected by 12-lead electrocardiography and warranting hospitalization and/or antiarrhythmic drugs. Congestive heart failure was defined as heart failure necessitating hospitalization, oxygenation, and diuretics. Peripheral artery disease was defined as arteriosclerosis obliterans necessitating intervention. Ischemic stroke was defined as stroke necessitating hospitalization and confirmed by head magnetic resonance imaging. 

- To confirm the advantages of the referral system over conventional follow-up, the data regarding risk factor control during the follow-up period would be helpful.

Thank you for your suggestion. Unfortunately, we could not obtain details regarding risk factor control for the conventional follow-up group, particularly for patients followed up at outpatient clinics. However, we have added the baseline and latest follow-up data for the unique referral system group in the Results section (lines 219-227).

- The study design is unclear. Particularly, methodology for data collection should be explained clearly. Data were collected every year or at the specific time point?

We apologize for the lack of clarity. Follow-up data were obtained when patients made regular outpatient visits to our hospital at annual referral visits following the Shizuoka IHD referral system, and at non-routine visits after referral from a general practitioner in relation to an event. In order to supplement the missing death events and non-cardiac events, we also performed a prognostic study by analyzing telephone and medical records obtained between March 2019 and May 2020. We have explained this in the Materials and Methods section (lines 136-140).

---

## [Decision Letter · Decision Letter 1]

15 Oct 2020

PONE-D-20-25225R1

Unique referral system contributes to long-term net clinical benefits in patients undergoing secondary prevention therapy after percutaneous coronary intervention

PLOS ONE

Dear Dr. KAGEYAMA,

Thank you for submitting your manuscript to PLOS ONE. After careful consideration, we feel that it has merit but does not fully meet PLOS ONE’s publication criteria as it currently stands. Therefore, we invite you to submit a revised version of the manuscript that addresses the points raised during the review process.

We look forward to receiving your revised manuscript.

Kind regards,

Yoshihiro Fukumoto

Academic Editor

PLOS ONE

Reviewers' comments:

Reviewer's Responses to Questions

**Comments to the Author**

1. If the authors have adequately addressed your comments raised in a previous round of review and you feel that this manuscript is now acceptable for publication, you may indicate that here to bypass the “Comments to the Author” section, enter your conflict of interest statement in the “Confidential to Editor” section, and submit your "Accept" recommendation.

Reviewer #1: (No Response)

Reviewer #2: (No Response)

2. Is the manuscript technically sound, and do the data support the conclusions?

Reviewer #1: Yes

Reviewer #2: Yes

3. Has the statistical analysis been performed appropriately and rigorously? 

Reviewer #1: Yes

Reviewer #2: Yes

4. Have the authors made all data underlying the findings in their manuscript fully available?

Reviewer #1: Yes

Reviewer #2: Yes

5. Is the manuscript presented in an intelligible fashion and written in standard English?

Reviewer #1: Yes

Reviewer #2: Yes

6. Review Comments to the Author

Reviewer #1: The authors almost responded to the reviewer’s remarks in detail. No additional comments would be needed.

Reviewer #2: I feel that the manuscript has been revised well.

Now, I have one comment.

- Please discuss why the clinical event occurrence was favorable in the referral system group, even though some risk factor control has got worse slightly.

7. PLOS authors have the option to publish the peer review history of their article (what does this mean?). If published, this will include your full peer review and any attached files.

Reviewer #1: No

Reviewer #2: No

---

## [Author Response · Author response to Decision Letter 1]

4 Nov 2020

Reviewers' comments:

Reviewer #1: The authors almost responded to the reviewer’s remarks in detail. No additional comments would be needed.

Response: Thank you for the positive feedback.

Reviewer #2: I feel that the manuscript has been revised well.

Response: Thank you for the positive feedback.

Comment 1: Now, I have one comment. - Please discuss why the clinical event occurrence was favorable in the referral system group, even though some risk factor control has got worse slightly.

Response: Thank you for your valuable comments. As you have correctly observed, risk factor control in the referral group only showed an improvement in LDL-C levels and exacerbation in HbA1c and blood pressure control compared to baseline. Despite the worsening of the latter two factors, the incidence of clinical events was lower in the referral group than in the conventional follow-up group. This discrepancy can be explained by the fact that our unique referral system lowered the threshold for general practitioners in recommending their patients to consult with a cardiologist. In turn, this could have potentially led to a net clinical benefit for the patient as it was easier to consult with other specialists in the general hospital, such as cerebrovascular disease and bleeding events specialists, using the referral to cardiologists as a starting point. Another potential reason is the fact that these factors do not perhaps pose the same level of risk of developing a clinical event. Many large clinical trials have reported that cardiovascular events are directly correlated with LDL-C levels in the secondary prevention of ischemic heart disease; however, it is possible that HbA1c levels or blood pressure are inferior risk indicators for cardiovascular events compared to LDL-C levels. In a large population-based prospective study in Japan, the risk of coronary artery disease was significantly higher in the hypertensive group (systolic blood pressure of 135 mm Hg or higher, diastolic blood pressure of 85 mm Hg or higher, or those taking antihypertensive drugs) than in a normotensive group. However, the risk probably could not have counteracted the impact of the improved prognostic effect of lower LDL-C in this study population. We have added these points to the Discussion section of the revised manuscript.

Discussion, p.26, lines 343-358:

“In this study, risk factor control in the referral group only showed an improvement in LDL-C levels and an exacerbation in HbA1c and blood pressure control compared to baseline. Despite the worsening of the latter two factors, the incidence of clinical events was still lower in the referral group than in the conventional follow-up group. This discrepancy can be explained by the fact that our unique referral system lowered the threshold for general practitioners in recommending their patients to consult with a cardiologist. In turn, this could have potentially led to a net clinical benefit for the patient since it was easier to consult with other specialists in the general hospital, such as cerebrovascular disease and bleeding event specialists, using the referral to the cardiologist as a starting point. Another potential reason is the fact that these factors perhaps do not pose the same level of risk of developing a clinical event. Many large clinical trials have reported that cardiovascular events are directly correlated with LDL-C levels in the secondary prevention of ischemic heart disease [4, 16]; however, it is possible that HbA1c levels or blood pressure are inferior risk indicators for cardiovascular events compared to LDL-C levels [19]. In a large population-based prospective study in Japan, the risk of coronary artery disease was significantly higher in the hypertensive group than in the normotensive group [20]. However, the risk probably could not have counteracted the impact of the improved prognostic effect of lower LDL-C in this study population.”

---

## [Decision Letter · Decision Letter 2]

9 Nov 2020

Unique referral system contributes to long-term net clinical benefits in patients undergoing secondary prevention therapy after percutaneous coronary intervention

PONE-D-20-25225R2

Dear Dr. KAGEYAMA,

We’re pleased to inform you that your manuscript has been judged scientifically suitable for publication and will be formally accepted for publication once it meets all outstanding technical requirements.

Kind regards,

Yoshihiro Fukumoto

Academic Editor

PLOS ONE

Additional Editor Comments (optional):

Reviewers' comments:

Reviewer's Responses to Questions

**Comments to the Author**

1. If the authors have adequately addressed your comments raised in a previous round of review and you feel that this manuscript is now acceptable for publication, you may indicate that here to bypass the “Comments to the Author” section, enter your conflict of interest statement in the “Confidential to Editor” section, and submit your "Accept" recommendation.

Reviewer #2: All comments have been addressed

2. Is the manuscript technically sound, and do the data support the conclusions?

Reviewer #2: Yes

3. Has the statistical analysis been performed appropriately and rigorously? 

Reviewer #2: Yes

4. Have the authors made all data underlying the findings in their manuscript fully available?

Reviewer #2: Yes

5. Is the manuscript presented in an intelligible fashion and written in standard English?

Reviewer #2: Yes

6. Review Comments to the Author

Reviewer #2: The manuscript has been revised well.

Now I have no further comments on this version of the manuscript.

7. PLOS authors have the option to publish the peer review history of their article (what does this mean?). If published, this will include your full peer review and any attached files.

Reviewer #2: No

---

## [Editor Report · Acceptance letter]

11 Nov 2020

PONE-D-20-25225R2 

Unique referral system contributes to long-term net clinical benefits in patients undergoing secondary prevention therapy after percutaneous coronary intervention 

Dear Dr. KAGEYAMA:

I'm pleased to inform you that your manuscript has been deemed suitable for publication in PLOS ONE. Congratulations! Your manuscript is now with our production department. 

Kind regards, 

on behalf of

Dr. Yoshihiro Fukumoto 

Academic Editor

PLOS ONE